# Attachment of Proteolytic Enzyme Inhibitors to Vascular Prosthesis—An Analysis of Binding and Antimicrobial Properties

**DOI:** 10.3390/molecules29050935

**Published:** 2024-02-21

**Authors:** Aleksandra Mordzińska-Rak, Katarzyna Szałapata, Jerzy Wydrych, Mariusz Gagoś, Anna Jarosz-Wilkołazka

**Affiliations:** 1Department of Biochemistry and Biotechnology, Institute of Biological Science, Maria Curie-Skłodowska University, Akademicka 19, 20-031 Lublin, Poland; aleksandra.mordzinska-rak@umlub.pl (A.M.-R.); katarzyna.szalapata@mail.umcs.pl (K.S.); 2Institute of Biological Sciences, Maria Curie-Skłodowska University, Akademicka 19, 20-031 Lublin, Poland; jerzy.wydrych@mail.umcs.pl; 3Department of Cell Biology, Institute of Biological Science, Maria Curie-Skłodowska University, Akademicka 19, 20-031 Lublin, Poland; mariusz.gagos@mail.umcs.pl

**Keywords:** vascular prosthesis, AEBSF, puromycin, *Staphylococcus aureus*, FTIR analysis

## Abstract

Prosthetic infections are associated with high morbidity, mortality, and relapse rates, making them still a serious problem for implantology. *Staphylococcus aureus* is one of the most common bacterial pathogens causing prosthetic infections. In response to the increasing rate of bacterial resistance to commonly used antibiotics, this work proposes a method for combating pathogenic microorganisms by modifying the surfaces of synthetic polymeric biomaterials using proteolytic enzyme inhibitors (serine protease inhibitors—4-(2-aminoethyl)benzenesulfonyl fluoride hydrochloride and puromycin). While using techniques based on the immobilization of biologically active molecules, it is important to monitor the changes occurring on the surface of the modified biomaterial, where spectroscopic techniques (e.g., FTIR) are ideal. ATR-FTIR measurements demonstrated that the immobilization of both inhibitors caused large structural changes on the surface of the tested vascular prostheses (polyester or polytetrafluoroethylene) and showed that they were covalently bonded to the surfaces of the biomaterials. Next, the bactericidal and antibiofilm activities of the tested serine protease inhibitors were determined using the CLSM microscopic technique with fluorescent staining. During LIVE/DEAD analyses, a significant decrease in the formation of *Staphylococcus aureus* biofilm after exposure to selected concentrations of native inhibitors (0.02–0.06 mg/mL for puromycin and 0.2–1 mg/mL for 4-(2-aminoethyl)benzenesulfonyl fluoride hydrochloride) was demonstrated.

## 1. Introduction

A biomaterial is now defined as a substance that has been engineered to take a form that, alone or as part of a complex system, is used for direct interactions with components of living systems in the course of any therapeutic or diagnostic procedure. Biomedical materials are used as medical devices (or components thereof) and are usually intended to be in long-term contact with biological tissues. Examples of biomedical materials are prostheses, reconstituted tissues, and intravenous catheters. The basic division of biomaterials used in implantology is based on their origin, and two main groups of biomaterials are distinguished. One group comprises natural protein- and polysaccharide-based biomaterials, and the other group contains biomaterials of synthetic origin, such as different polymers, ceramics, metals, and composites [1]. Polymers are the most popular synthetic materials because of their high levels of biodegradability, biocompatibility, and mechanical strength. Moreover, polymers are highly flexible and have a positive effect on cell adhesion [2]. Unfortunately, despite the many benefits of using all available biomaterials in implantology, many studies indicate frequent infections of these biomaterials, even before they are implanted into the human body. Gram-positive bacteria from the *Staphylococcus* genus, Gram-negative bacteria such as *Escherichia coli* and *Pseudomonas aeruginosa*, and fungi from the *Candida* genus are responsible for the infections [3]. This serious and common problem prevents the proper functioning of a newly implanted prosthesis. The functionalization of biomaterials with different active substances gives new properties to the prostheses used. One of the most frequent modifications is the introduction of changes in the structure of biomaterials with heparin or albumin and substances that have antimicrobial activity (e.g., antibiotics, silver nanoparticles) or the finishing of the biomaterial surface using repellent coatings [4,5,6]. An example of new substances with antimicrobial potential used in regenerative medicine may be inhibitors of proteolytic enzymes. Since proteases are identified as virulence factors of many pathogenic microorganisms, using proteolytic enzyme inhibitors as antimicrobial substances seems to be the most reasonable action [7,8,9]. Considering the multifunctionality of trypsin inhibitors, they are excellent candidates for studies focused on the assessment of their antimicrobial potential after their immobilization on different biomaterials. Proteases, e.g., trypsin, are ubiquitous among prokaryotic and eukaryotic cells and function in several metabolic processes [10]. In our previous paper [11], we identified the bacteriostatic and bactericidal effects of a synthetic inhibitor of the serine-type protease 4-(2-aminoethyl) benzenesulfonyl fluoride hydrochloride (AEBSF) on *P. aeruginosa*, *E. coli*, and *Staphylococcus aureus*.

Given the highlighted points and considering the search for new alternatives for treating microbial infections of biomaterials, the present work aimed to study (1) the interactions between trypsin (a serine-type proteolytic enzyme) and enzymatic inhibitors (puromycin and AEBSF) and (2) the antibacterial activity of selected serine-type protease inhibitors in contact with biomaterials for the prevention of bacterial biofilm formation. The immobilization of an active substance on the biomaterial surface affects its activity and can impart new properties to the modified surface. This work focuses on the structural analysis of two synthetic vascular prostheses made of polyester or polytetrafluoroethylene, differing in the presence of polyester groups and an additional protein layer (collagen or gelatin). ATR-FTIR spectroscopy was used to check trypsin–inhibitor interactions and confirm the process of immobilization of the tested inhibitors on the vascular prosthesis surface.

## 2. Results and Discussion

The research problem was addressed comprehensively in this study; in the first stage, the thesis that proteolytic enzyme inhibitors attach to the proteolytic enzyme molecule and modify its active center was checked and confirmed using ATR-FTIR spectroscopy. Following the tests, two types of vascular prostheses were used, and the covalent immobilization of the selected inhibitors on their surfaces was carried out to confirm their attachment also in the context of changes in the structure of polymeric materials.

### 2.1. ATR-FTIR Analysis of Biomolecular Mechanisms of Inhibitor Binding to Proteolytic Enzyme

The first stage of the research involved an analysis of the native enzyme–inhibitor interaction using ATR-FTIR spectroscopy. It was checked whether the addition of the inhibitor changes the structure of the enzyme molecule, and the functional groups involved in the binding between the two molecules were analyzed using the ATR-FTIR method (Figure 1).

In the TRYP solution, the amide was an essential protein group marker. All the spectra shown above fit a typical infrared protein profile. In the 1800–1300 cm^−1^ region, amide I and amide II bands appeared as the most prominent features (Table 1). These bands corresponded to the collective vibrations of the peptide chain backbone [12]. The most intense absorption band in the protein was due to the presence of the amide I group in the region of 1680–1631 cm^−1^, which is primarily governed by the stretching vibrations of C=O (70–85%) and C-N bonds (10–20%) [13]. In addition, the amide I group resulted from the collective C=O stretching vibrations of the protein chain, with a small contribution from the in-plane δNH bending mode [12]. The amide I section of the spectra was also reported to include the region of 1648–1657 cm^−1^, assigned to the α-helix, and the region of 1610–1640 cm^−1^, assigned to the β-helix. The amide II group (1580–1480 cm^−1^) (C-N stretching bond coupled with N-H bending modes) [14] corresponded to a combination of the in-plane δNH bending mode with the stretching of the C-N peptide bond [15]. The peak in the wavenumber range of 1472–1239 cm^−1^ corresponded to the N-H bending bond and the C-N stretching vibrations of the amide III group; it was usually another combination of the N-H and C-N vibrations and was a less intense group than amide I and amide II. As can be seen, after mixing TRYP and PUR, the absorbance of the amide I group decreased and was shifted toward lower frequencies by 1 cm^−1^ compared to TRYP (Figure 1A, line a). Surprisingly, the amide II region was shifted toward higher frequencies by 5 cm^−1^ after the addition of PUR (Figure 1A, line b). This effect was related to the interactions of the N-H bond originating from the amide II group, which may mean that PUR induced changes in the secondary structure of TRYP that led to enzyme inactivation. In addition, the ratio of the amide I to amide II group in the mixture of the TRYP and PUR spectra was also changed.

An additional band (at 1656 cm^−1^) from the vibrations of the aromatic group in the PUR molecule appeared in addition to the shifts observed in the amide bands. It was described earlier that the phenomenon of amide band shifting also occurs when ordinary deionized water is added to the TRYP molecule [16]. Therefore, it is worthwhile to determine whether the phenomenon of shifting is indeed due to interactions between the compounds or whether it is merely a meaningless change caused by the appearance of moisture in the studied material. Since deionized water caused significant shifts in the maximum absorption of the amide I group and the shift observed in our study was insignificant, amounting to only 1 cm^−1^, it can be assumed that it appeared only due to the interactions occurring between the enzyme and the inhibitor molecule.

A slightly different situation was observed in the case of the mixture of TRYP and AEBSF. Figure 1 B shows the spectra of TRYP alone (line a), the mixture of TRYP + AEBSF (line b), and AEBSF (line c). As can be seen, no spectral shift of the band related to the amide I region and no changes in the shape of this spectrum were observed. The changes in the region related to the amide II group may be due to the overlapping of the band from AEBSF (green line) with the amide band.

Therefore, it may be concluded that the nature of the interactions between TRYP and AEBSF is not obvious, and they do not induce changes in the secondary structure of the enzyme as much as in the case of PUR.

Despite the lack of a conclusive answer confirming that AEBSF significantly alters the structure of the active site of trypsin, it is worth noting that the observed 1 cm^−1^ shift of the amide II band toward lower frequencies may contribute to the elucidation of the interaction between these two molecules. Furthermore, the addition of the AEBSF inhibitor resulted in the appearance of a new band at 1599 cm^−1^, further supporting our hypothesis. The specific serine protease inhibitor AEBSF inhibited the penetration of the basement membrane by porcine pseudorabies virus by 88.1% [17] and displayed bacteriostatic and bactericidal effects on *P. aeruginosa*, *E. coli*, and *S. aureus* [11].

### 2.2. ATR-FTIR Analysis of Structural Changes after Immobilization of Tested Inhibitors on ePTFE Prosthesis

Figure 2 presents the ATR-FTIR spectra of the control ePTFE prosthesis (washed with phosphate buffer) (black line), the ePTFE prosthesis functionalized with GLA (5%) only (green line), the ePTFE prosthesis functionalized with GLA (5%) and AEBSF (0.6 mg/mL) (red line), and AEBSF alone (0.6 mg/mL) (blue line). Figure 3 shows the ATR-FTIR spectra of the control ePTFE prosthesis (washed with phosphate buffer) (black line), the ePTFE prosthesis functionalized with GLA (5%) only (green line), the ePTFE prosthesis functionalized with GLA (5%) and PUR (0.04 mg/mL) (red line), and PUR alone (0.04 mg/mL) (blue line). Strictly defined concentrations of the inhibitors (PUR and AEBSF) having the highest activity after immobilization on the ePTFE prosthesis were selected for this analysis. To enhance the spectral differences, inverted second derivatives of the aforementioned spectra were calculated (Figure 2a–c and Figure 3a–c). Based on the inverted second derivatives, spectral assignments were performed, as presented in Table 2.

The absorption band shifts and intensity differences associated with GLA, AEBSF, and PUR immobilization on the ePTFE prosthesis were analyzed and compared with the control ePTFE prosthesis based on the total peak areas calculated for the inverted second derivatives of the ATR-FTIR spectra.

In the spectrum of the ePTFE control prosthesis, we observed the stretching vibrations of the -CH group at 2956 cm^−1^, the symmetric vibrations of the aliphatic -CH_2_ group at 2914 cm^−1^, and the stretching vibrations of the -CH_2_ group at 2846 cm^−1^ [18]. The ePTFE prosthesis was gelatin-coated; therefore, in the spectrum, we observed an amide I group (1624 cm^−1^) as a band from the β-sheet with cleavage to the α-helix (1654 cm^−1^) and an amide II group (1546 cm^−1^). The absorption band at 1447 cm^−1^ was related to the deformation vibrations of the -CH_2_ group of polytetrafluoroethylene. The band at 1397 cm^−1^ was related to the deformation vibrations of the -CH_3_ group. The maximum absorption at 1234 cm^−1^ originated from the fluoride group -CF building the ePTFE prosthesis structure [19]. In addition, the absorption maxima at 1198 cm^−1^ and 1145 cm^−1^ were characteristic of the asymmetric and symmetric stretching vibrations, respectively, of the -CF_2_ group present in the ePTFE prosthesis [20,21]. Upon the immobilization of 5% GLA on the surface of the ePTFE prosthesis (Figure 2), a decrease in the intensity of the amide I and II groups and a spectral shift of both bands corresponding to the α-helix (by 4 cm^−1^) and the β-helix (by 3 cm^−1^) toward higher frequencies were observed. The absorption band maximum coming from the deformation vibrations of the -CH_2_ group was shifted toward higher frequencies (by 7 cm^−1^) in comparison to the control ePTFE prosthesis. The absorption band derived from the vibrations of the -CH_3_ group was characterized by a 6 cm^−1^ shift toward higher frequencies. No change in the absorption maximum was observed for the vibrations of the -CF group compared to the control ePTFE prosthesis. The absorption bands coming from the symmetric stretching vibrations of the -CF_2_ group were characterized by an 8 cm^−1^ shift in the absorption maxima toward higher frequencies. It is also worth noting that there were changes in the absorption bands of the stretching vibrations of the -CH group with the band intensity at 2956 cm^−1^. Additionally, the ratio of the bands at 2956 cm^−1^ and 2921 cm^−1^ in the case of ePTFE + GLA was also changed in comparison to the control ePTFE prosthesis. The results described above demonstrate the interaction of the -COOH group derived from GLA with the -NH bond (from the amide II group) of gelatin coating the prosthesis (Figure 2).

AEBSF immobilization on the ePTFE prosthesis also induced changes in the structure of this biomaterial. The inhibitor addition caused a 3 cm^−1^ spectral shift in the absorption maximum of the band originating from the vibrations of the -CH group toward higher frequencies (in comparison to the control prosthesis). The ratio of the bands at 2959 cm^−1^ and 2921 cm^−1^ was also changed. After AEBSF immobilization, a greater ratio difference between 2959 cm^−1^ and 2921 cm^−1^ was observed. The addition of the AEBSF inhibitor resulted in a 3 cm^−1^ shift in the amide II group toward higher frequencies compared to the spectrum of the GLA-modified ePTFE prosthesis. In addition, a 4 cm^−1^ shift in the absorption maxima for the vibrations of the -CF group toward lower values was observed. The absorption spectra from the asymmetric stretching vibrations of the -CF_2_ group were characterized by shifts in the absorption maxima toward higher frequencies (by 2 cm^−1^) and a 4 cm^−1^ shift for symmetric vibrations compared to the spectrum of the GLA-modified prosthesis. The shifts in the absorption bands described above are indicative of the attachment of the -NH_2_ group of AEBSF to the free end of the GLA molecule (-CHO group) and, in part, the attachment of the -NH group of gelatin to the ePTFE prosthesis itself.

The process of PUR immobilization on the ePTFE prosthesis was confirmed after the analysis of absorption band shifts and their intensity differences (Figure 3). After the immobilization of PUR, a 2 cm^−1^ shift in the absorption maxima of the band originating from the stretching vibrations of the -CH group toward lower frequencies (relative to the ePTFE control) was observed. Here, slight differences in the ratio of the 2954 cm^−1^ and 2921 cm^−1^ bands were observed in comparison to the control sample. The addition of PUR also shifted the amide I group (by 5 cm^−1^) and the amide II group (by 7 cm^−1^) toward lower frequencies relative to the spectrum of the GLA-modified ePTFE prosthesis. In addition, the cleavage of the amide I group disappeared. The absorption band originating from the asymmetric stretching vibrations of the -CF_2_ group was characterized by a shift in the absorption maxima toward higher values (by 3 cm^−1^); similarly, the band for the symmetric stretching vibrations of the -CF_2_ group was shifted by 8 cm^−1^ toward higher values. The observed shifts in the absorption bands are indicative of the attachment of the -NH_2_ group of the PUR molecule to the free end of the GLA molecule (to the -CHO group) and, in part, the attachment of the -NH group of gelatin to the ePTFE prosthesis itself (Figure 3).

The ePTFE (polytetrafluoroethylene) graft is a fluorinated polyethylene with the formula (CF_2_-CF_2_)*_n_* [22]. It is coated with gelatin, which causes amide bands from this protein to be observed in FTIR spectral analyses. Similar studies were performed by another research team using caffeic acid to immobilize gelatin on another support, i.e., bioactive glass. The resulting amide bands were observed using infrared FTIR spectroscopy [23]. It is likely that in the immobilization process (through binding to the -CHO group) carried out using GLA as a linker molecule, amide groups are primarily involved in the attachment of the crosslinking compound. The second -CHO group of GLA binds to the inhibitor molecule by attaching to its -NH_2_ group, while the -CO group of PUR also partially binds to the amide group of gelatin. The numerous shifts in the absorption maxima of the vibrations of the -CH, -CH_2_, and -CH_2_CH_3_ groups, the amide groups, and the -CF and -CF*_2_* groups indicate the correct process of inhibitor immobilization by the crosslinker on the ePTFE vascular prosthesis.

### 2.3. ATR-FTIR Analysis of Structural Changes after Immobilization of Tested Inhibitors on HEM Prosthesis

Figure 4 presents the ATR-FTIR spectra of the control HEM prosthesis (washed with phosphate buffer) (black line), the HEM prosthesis functionalized with GLA (5%) only (green line), the HEM prosthesis functionalized with GLA (5%) and AEBSF (0.6 mg/mL) (red line), and AEBSF alone (0.6 mg/mL) (blue line). Figure 5 presents the ATR-FTIR spectra of the control HEM prosthesis (washed with phosphate buffer) (black line), the HEM prosthesis functionalized with GLA (5%) only (green line), the HEM prosthesis functionalized with GLA (5%) and PUR (0.04 mg/mL) (red line), and PUR alone (0.04 mg/mL). These concentrations of AEBSF and PUR were used because they caused the highest inhibitory activity after their immobilization on the HEM prosthesis. To enhance the spectral differences, inverted second derivatives of the aforementioned spectra were calculated and are presented in Figure 4a–c and Figure 5a–c. Based on the inverted second derivatives, the spectral assignments were performed, as presented in Table 3.

The absorption band shifts and intensity differences associated with GLA, AEBSF, and PUR immobilization on the HEM prosthesis were analyzed and compared with the control HEM prosthesis based on the total peak areas calculated for the inverted second derivatives of the ATR-FTIR spectra.

In the spectrum of the HEM control prosthesis, the stretching vibrations of the -CH group at 2958 cm^−1^ [24], the symmetric vibrations of the aliphatic -CH_2_ group at 2921 cm^−1^, and the stretching vibrations of the -CH_2_ group at 2849 cm^−1^ were observed. The HEM prosthesis was coated with collagen; hence, the spectrum showed the amide I group split into an α-helix (1656 cm^−1^) and a β-sheet (1627 cm^−1^) and the amide II group (1549 cm^−1^) [25]. The absorption band at 1452 cm^−1^ originated from the deformation vibrations of the -CH group of the polyester. In addition, the band at 1404 cm^−1^ originated from the -OH group of the carboxyl moiety of polyester. The absorption bands with maxima at 1238 cm^−1^, 1198 cm^−1^, 1161 cm^−1^, and 1082 cm^−1^ came from the carbonyl group -CO building up the structure of the polyester skeleton of the HEM prosthesis [20]. During GLA immobilization on the HEM prosthesis (Figure 4, green line), there was a significant reduction in the intensity of the amide I and amide II groups and shifts toward higher frequencies in the amide I group corresponding to the α-helix (by 7 cm^−1^) and amide I corresponding to the β-sheet (by 5 cm^−1^) relative to the control sample. The absorption maximum of the band originating from the vibrations of the amide II group and the vibrations of the -OH and -CH groups did not change after GLA immobilization. However, changes were observed (9 cm^−1^ shift toward higher frequencies) in the maximum absorption of the carbonyl group -CO vibrations originating from the backbone of the polyester chain in comparison to the control. The absorption maximum of the vibrations of the -CH group also changed and exhibited a 2 cm^−1^ shift toward higher frequencies in comparison to the control. The above-described results demonstrate the attachment of the -CHO group of GLA to the N-H bond (amide II) of the collagen layer of the HEM prosthesis.

AEBSF immobilization on the HEM prosthesis also significantly affected changes in the biomaterial’s structure. The addition of this inhibitor caused a 3 cm^−1^ shift in the absorption maxima of the band originating from the stretching vibrations of the -CH group and a significant increase in the intensity of this band relative to the control. In addition, the change in the band originating from the vibrations of the -CH_2_ group consisted of a 3 cm^−1^ shift in the absorption maxima toward higher values relative to the control HEM prosthesis. The addition of the AEBSF inhibitor also shifted the band of amide I and altered the absorption maxima originating from the α-helix and β-sheet (7 cm^−1^ and 5 cm^−1^ shifts, respectively) toward lower frequencies in comparison to the spectrum of the GLA-modified HEM prosthesis. In addition, the band originating from the deformation vibrations of the -CH group also changed, as there was a 4 cm^−1^ shift in the absorption maximum toward lower values. The vibrations of the -OH group were also shifted by 4 cm^−1^ toward higher values in comparison to the control HEM prosthesis. Changes were also observed in the vibrations of the -CO group (absorption band at 1165 cm^−1^), whose absorption maxima were shifted by 5 cm^−1^ toward lower values in comparison to the spectrum of the GLA-modified HEM prosthesis.

The process of PUR immobilization on the HEM prosthesis was confirmed after the analysis of the shifts of the absorption bands originating from the vibrations of the different functional groups present in the biomaterial. After the immobilization of PUR, a 7 cm^−1^ shift in the absorption maxima of the band originating from the symmetric vibrations of the -CH_2_ group toward higher frequencies in comparison to the absorption band originating from the spectrum of the GLA-modified HEM prosthesis was observed. The maximum at 2922 cm^−1^ related to the symmetric vibrations of the -CH_2_ group almost returned to the value observed in the control. The amide I group did not change, while a 3 cm^−1^ shift in the absorption maxima of the amide II group toward higher frequencies was observed. The addition of the inhibitor also caused a 2 cm^−1^ shift in the absorption maxima of the band originating from the vibrations of the -OH group toward higher frequencies relative to the control. Changes were also observed in the band originating from the vibrations of the -CO group (absorption band at 1163 cm^−1^), with a 7 cm^−1^ shift in the absorption maximum of the band toward lower values compared to the absorption spectrum of the GLA-modified HEM prosthesis.

The HEM prosthesis is a commercially available collagen-coated vascular polyester prosthesis; it can also be supplied in a version with additionally immobilized heparin, as described in [26]. Collagen-derived amide bonds involved in the binding of GLA during the immobilization process were observed in the case of this prosthesis. In contrast, the inhibitor molecules (PUR and AEFBS) bind both to the -CHO group of GLA and to the -C=O group of the polyester backbone. After the immobilization of AEBSF or the complete “capture” of the carbonyl group by the PUR molecule, the difference in the ratio of the absorption maxima of the carbonyl group bond vibrations was observed compared to the amide I group.

### 2.4. CLSM Analysis of PUR and AEFBS Antimicrobial Activity (Live/Dead Analysis)

*Staphylococcus aureus* is one of the most common and opportunistic pathogens responsible for infections in public health [27,28]. Implant infections caused by this bacterium carry a high risk, especially for older patients and those with various metabolic burdens [29]. Due to the widespread phenomenon of antibiotic resistance among pathogenic microorganisms, increasing emphasis is placed on the search for alternative substances with equally high bactericidal and bacteriostatic potential, e.g., copper–silver alloys [30], derivatives of diindolylomethane [31], microbial proteolytic enzymes [32], PLGA/xylitol nanoparticles [33], unsaturated C18 fatty acids [34], or small-molecule inhibitors like savarin [35]. Moreover, research conducted on bactericidal and antibiofilm activities requires the use of appropriate measurement methods. The most popular and cheapest are tests based on the use of uncoated polystyrene plates with colored substances (including crystal violet) coupled with absorbance measurements on microplate readers [36]. Techniques based on light microscopy and scanning electron microscopy (SEM) [31] are also often used to assess the effectiveness of antimicrobial and antibiofilm effects. However, one of the most common methods providing plenty of valuable data is confocal laser scanning microscopy (CLSM) [37]. CLSM allows the full imaging of the 3D structure of the bacterial biofilm, the observation of the localization of individual biofilm components (proteins, polysaccharide matrix, bacteria cells), and the study of the process of biofilm formation during a specified time, even in flow systems [38,39,40].

In this work, the antimicrobial activity of the tested inhibitors (PUR and AEBSF) against the model strain of *Staphylococcus aureus* was tested using CLSM analysis. For this purpose, the bacterial biofilm formed on the surface of the HEM prosthesis was exposed to various concentrations of PUR and AEBSF (the concentrations were selected based on the analyses of their activity after their immobilization on the prosthesis [11] for 3 h); next, CLSM with live/dead staining was used. The LIVE/DEAD analysis was performed to observe the effect of the native inhibitor concentrations (PUR and AEBSF) on the bactericidal properties of biomaterials functionalized with 5% GLA. Crystal violet, which readily penetrates the cell wall and the cytoplasmic membrane of Gram-positive bacteria, can also be used to perform such tests [36].

Differences were observed in the fluorescence intensity of dead *S. aureus* cells after their exposure to the selected concentrations of PUR and AEBSF compared to the controls (Figure 6). The use of PUR at a concentration of 0.02 mg/mL caused the death of only some of the bacterial cells (red fluorescence), and its rate increased with the use of a higher antibiotic dose. As observed in the confocal microscope images, PUR at a dose of 0.06 mg/mL caused the death of a large number of cells, but a relatively large number of live cells were still observed. In the case of AEBSF, the effect was much more pronounced, and the use of the lowest dose (0.2 mg/mL) already contributed to a higher cell death rate than when PUR was added at a concentration of 0.02 mg/mL. Furthermore, the use of AEBSF at a concentration of 1 mg/mL resulted in the death of a significant number of cells immobilized on the prosthesis. It was demonstrated that the higher concentrations of both inhibitors caused the death of a greater number of *Staphylococcus aureus* cells, thus confirming their antibacterial properties.

The second stage of the experiment was performed to compare the qualitative analysis based on the fluorescence intensity of microscopic photos with the quantitative analysis based on counting live and dead *S. aureus* cells. The quantitative analysis of *S. aureus* cell death after 3 h of exposure to native PUR and AEBSF molecules indicated a dose-dependent effect (Figure 7A,B). Based on CLSM images, calculations showed 11.9% *S. aureus* dead cells at the lowest PUR concentration (0.02 mg/mL) and 30.52% dead cells at the highest one (0.06 mg/mL). The obtained values indicated an almost 3-fold increase in the number of dead bacterial cells with a 3-fold increase in PUR concentration. Similar dependencies were observed for the synthetic AEBSF inhibitor. At the lowest AEBSF concentration (0.2 mg/mL), 21.08% dead *S. aureus* cells were counted, while at the highest concentration (1 mg/mL), it was as much as 66.3%. This also indicated a dose-dependent effect of this inhibitor. However, high dead cell numbers were observed at higher final concentrations of AEBSF compared to PUR concentration. It is worth emphasizing that a 66.3% mortality of *S. aureus* cells forming a biofilm on the surface of the prosthesis was confirmed for the AEBSF inhibitor at half of the MIC value (2 mg/mL), as determined in an earlier study [11].

### 2.5. Analysis of S. aureus Growth Inhibition in Suspension Cultures

The results illustrating the inhibition of *S. aureus* growth in suspension cultures are presented in Figure 8.

Using native and immobilized preparations of proteolytic enzyme inhibitors slowed down the cell proliferation of this opportunistic pathogen compared to the control growth sample (Figure 8). Almost all growth curves were characterized by a lower optical density than the control. The exception was the sample incubated with a fragment of the control HEM prosthesis (Figure 8A), where after 5 h of incubation, there was a significant increase in optical density. This phenomenon was most likely caused by the accelerated multiplication of *S. aureus* cells as a result of the initiation of the digestion of the unmodified gelatin coating on the biomaterial surface. The most effective modifications were the HEM prosthesis functionalized with 1 mg/mL of AEBSF (Figure 8B) and the HEM prosthesis functionalized with 0.04 mg/mL of PUR (Figure 8C). High growth inhibition was observed during incubations of *S. aureus* with solutions of native inhibitors—AEBSF at concentrations of 0.6 mg/mL and 1 mg/mL (Figure 8B) and 0.06 mg/mL of PUR (Figure 8C).

To confirm the results regarding the density of *S. aureus* suspension cultures, the quantitative measurement of cells was performed after 5 h of incubation (Table 4). The obtained results indicate the same order of magnitude of measurement [CFU/mL]. However, almost all of the tested variants of native and immobilized inhibitors (except HEM + AEBSF 0.2 mg/mL) showed a lower number of colony-forming units after 5 h of incubation compared to the growth of control *S. aureus* cultures and the sample incubated with the unmodified HEM prosthesis.

## 3. Materials and Methods

### 3.1. Enzymes and Inhibitors

Puromycin (PUR)—a natural aminonucleoside antibiotic (Figure 9a) with a molecular weight of 544.4 Da isolated from *Streptomyces alboniger* [41]. PUR inhibits protein synthesis by the induction of premature chain termination by acting as an analog of the 3′-terminal ends of aminoacyl-tRNA. Additionally, PUR is a reversible inhibitor of dipeptidyl-peptidase II (serine peptidase) and cytosol alanyl aminopeptidase (metallopeptidase). PUR is active against Gram-positive bacteria, less active against acid-fast bacilli, and more weakly active against Gram-negative microorganisms [42].4-(2-Aminoethyl)benzenesulfonyl fluoride hydrochloride (AEBSF)—a water-soluble, irreversible serine protease synthetic inhibitor (Figure 9b) with a molecular weight of 239.5 Da. AEBSF inhibits proteases like chymotrypsin, kallikrein, plasmin, thrombin, and trypsin [43].Trypsin (TRYP)—a model serine protease cleaving peptides on the C-terminal side of lysine and arginine residues. TRYP is produced in pancreatic acinar cells in an inactive form (known as trypsinogen) and is activated only in the lumen of the small intestine to digest proteins [44].

### 3.2. Biomaterials Used for Immobilization

The Hemagard Intergard (HEM) prosthesis is a collagen-coated knitted vascular polyester prosthesis (Getinge, Gothenburg, Sweden). This synthetic material consists of polyester fibers produced by the polycondensation of dicarboxylic acids with polyhydroxyl alcohols, most often terephthalic acid with ethylene glycerol. The HEM prosthesis is modified by the crosslinking of bovine collagen on the surface. This biomaterial is produced by using the reverse locknit knitting technique, which ensures long-term stability, tear strength, and tensile strength (https://www2.getinge.com/us/products/hospital/vascular-surgery-solutions/ (accessed on 18 February 2024)).

The polytetrafluoroethylene (ePTFE) prosthesis is a gelatin-coated ultrathin vascular prosthesis (Vascutek Terumo, Inchinnan, UK) produced by tetrafluoroethylene polymerization. In the structure of this synthetic material, there are carbon–fluorine and carbon–carbon bonds, which increase its resistance to high temperatures. In addition, this material is highly biocompatible, which facilitates its use, for example, in artificial tendons [45].

### 3.3. Analysis of Enzyme–Inhibitor Interactions Using ATR-FTIR Spectroscopy

The analyses were performed for TRYP (serine protease) by adding one of the tested inhibitors—synthetic AEBSF or natural PUR. Initially, the inhibitors and enzyme solutions were dissolved in separate Eppendorf tubes in 10 µL of water in the following amounts: TRYP—0.2 mg; AEBSF—2 mg; and PUR—4 mg. Next, each solution was applied separately to the ATR crystal, dried with nitrogen for 5 min, and then placed under a vacuum pump for 10 min. After this time, analyses were performed in two repetitions. Initially, separate absorption spectra of TRYP, AEBSF, or PUR and then the enzyme–inhibitor systems in a 1:1 molar ratio (TRYP-AEBSF, TRYP-PUR) were measured.

### 3.4. Covalent Immobilization of Protease Inhibitors on Vascular Prosthesis

The process of AEBSF and PUR immobilization was carried out in a 24-well plate. Fragments of the tested biomaterial (0.5 cm × 0.5 cm) were washed with distilled water and 0.1 M phosphate buffer pH 7.0. Next, the biomaterial surface activation process was started with the use of glutaraldehyde. The biomaterial was suspended in a solution of 5% glutaraldehyde (GLA) in 0.1 M phosphate buffer pH 7.0 and shaken for 1 h (75 RPM) at room temperature. Then, the biomaterial fragment was washed with 0.1 M phosphate buffer pH 7.0 until the characteristic smell of aldehyde disappeared, reduced with a solution of NaBH_4_ (for 15 min at 4 °C), and again washed with cold 0.1 M phosphate buffer pH 7.0. Then, the activated fragment of the biomaterial was placed in the inhibitor solution with an optimized concentration (5 mL of the inhibitor solution per 1 cm^2^ of the biomaterial with the final concentrations of PUR of 0.04 mg/mL and AEBSF of 0.6 mg/mL) and shaken for 3 h at room temperature [11]. Finally, the preparation was stored at 4 °C for 12 h to stabilize the modified surface. After this time, non-covalently bound inhibitor molecules were removed by washing with the following solutions with different pH and ionic strength values: 0.1 M phosphate buffer pH 7.0, 0.5 M NaCl, 0.1 M phosphate-citrate buffer pH 5, and distilled water. The active groups remaining on the carriers were blocked by washing with 0.5 M Tris-HCl buffer (pH 7.5). The functionalized vascular prostheses with immobilized inhibitor preparations were intensively rinsed with 0.1 M phosphate buffer pH 7.0.

For further analysis, the pieces of functionalized prostheses were placed in a desiccator for 48 h to dry thoroughly and stored in closed Eppendorf tubes afterward.

### 3.5. Analysis of Biomaterial Surface Structure Using ATR-FTIR Spectroscopy

Attenuated Total Reflection Fourier Transform Infrared Spectroscopy (ATR-FTIR) was used to study the influence of the presence of the inhibitors on the surfaces of the tested vascular prostheses. Fragments of vascular prostheses with the inhibitor immobilized on their surfaces were placed on a ZnSe crystal and dried under a nitrogen stream for about 5 min. ATR-FTIR spectra in the range of 4000–400 cm^−1^ were measured for the vascular prostheses using an FTIR VERTEX 70 (Bruker Optik GmbH, Ettlingen, Germany) spectrometer with an MCT detector. The spectral resolution was set at 4 cm^−1^, 8 cm^−1^, or 16 cm^−1^, which was related to the type of sample measured. Sixteen or thirty-two scans per sample were collected. OPUS 7.5, Grams/32 Version 4.0, and Grapher 8 were used to set the baseline, cut out unnecessary fragments of the analysis, and smooth the peaks.

### 3.6. Biofilm Formation on GLA-Activated HEM Prostheses

Fragments of the HEM vascular prosthesis activated with 5% GLA were inoculated with *Staphylococcus aureus* (ATCC 25923) with an optical density of 0.1 at 600 nm. Biofilms were cultured in tryptic soy broth (TSB) medium for 48 h (37 °C, 100 RPM). After this time, the fragments of the prosthesis were gently washed three times in sterile phosphate-buffered saline (PBS) and placed in fresh TSB medium with PUR (0.02–0.06 mg/mL) or AEBSF (0.2–1.0 mg/mL) for 3 h (37 °C, 100 RPM). After that, the HEM prostheses were washed three times in PBS before microscopic analysis.

### 3.7. Confocal Laser Scanning Microscopy Analysis (CLSM)

An Axiovert confocal microscope with a scanning head (LSM 5 PASCAL; Carl Zeiss, Jena, Germany) was used to determine the presence of biofilms qualitatively. The analysis was performed using a laser scanning confocal imaging system. The excitation/emission was 488 nm/<550 nm for SYTO 9 and 555 nm/>550 nm for propidium iodide, i.e., dyes staining nucleic acids in bacterial cells. A working solution of fluorescent stains was prepared by adding 3 µL of SYTO 9 and 3 µL of propidium iodide to 1 mL of filter-sterilized water, and 200 µL of the staining solution was added to the HEM vascular prosthesis. Next, the samples were incubated for 30 min in darkness at RT. Photos were taken at 400× zoom for AEBSF 1 mg/mL and for PUR 0.02 mg/mL; the rest were taken at 200× zoom.

Then, a quantitative analysis was performed on three independently taken photos for each variant and inhibitor concentration (3 repetitions). Cells were counted using ImageJ 1.53 t, and the results are presented in bar charts as mean values ± SDs (GraphPad Prism 8). All cells observed in a single photo were set to a 100% value (total bacteria cells), and then the number of cells with red fluorescence was calculated (dead bacteria cells). This number is presented as a percentage (relative to the entire number of cells visible in the photo) and constitutes the percentage of cell mortality in each research variant.

### 3.8. Analysis of S. aureus Growth Inhibition in Suspension Cultures

The Hemagard prosthesis fragments (1 cm × 1 cm) were modified according to the procedure described in Section 3.4. Additionally, control variants of unmodified and GLA-activated biomaterial were prepared, and as a positive control, a 1 mM gentamicin solution was used. The same concentrations of PUR (0.02, 0.04, 0.06 mg/mL) and AEBSF (0.2, 0.6, 1 mg/mL) were used in the experiment for modified biomaterials and native molecules.

Based on overnight culture, a *Staphylococcus aureus* (ATCC 25923) suspension was prepared in Mueller–Hinton Broth with an optical density of OD_600_ = 0.01. Then, 5 mL of the bacterial suspension and a fragment of the prosthesis (in the case of bacteria–biomaterial interactions) or 100 µL of native molecules (bacteria–native substance interactions) were added to Fischer tubes. The test tubes were placed in a laboratory shaker and incubated for 5 h (37 °C, 140 RPM). The optical density OD_600_ for each tested variant was measured every hour (bioSan DEN-600 photometer, Rīga, Latvia). Additionally, after 5 h of incubation, colonies were counted in each variant and presented in CFU/mL units. The experiment was performed in 3 independent repetitions.

## 4. Conclusions

The process of the immobilization of the tested low-molecular-weight molecules—proteolytic enzyme inhibitors—on polymeric vascular prostheses resulted in numerous structural changes in these polymeric materials, as observed using ATR-FTIR spectroscopy. The largest spectral shifts related to molecular interactions were observed for the amide groups, which are part of the compounds coating both polymeric prostheses. The LIVE/DEAD analyses using confocal microscopy confirmed the antibacterial effect of the tested inhibitors, i.e., their activity against *Staphylococcus aureus*, and highlighted the higher efficiency of AEBSF, which is a water-soluble, irreversible, synthetic serine protease inhibitor.

## Figures and Tables

**Figure 1 molecules-29-00935-f001:**
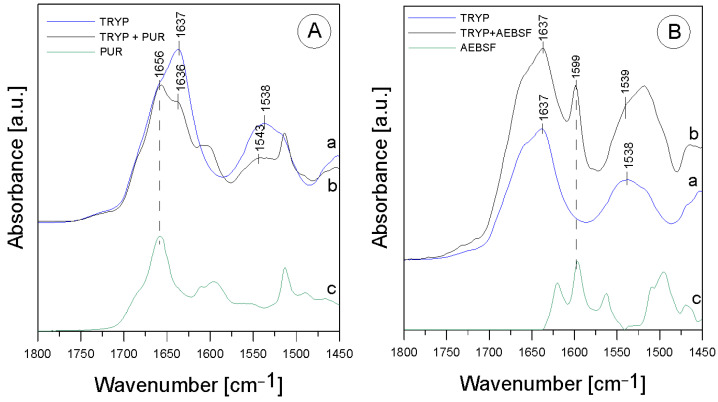
ATR-FTIR spectra for (**A**)—TRYP (a), TRYP + PUR (b), PUR (c); (**B**)—TRYP (a), TRYP + AEBSF (b), AEBSF (c).

**Figure 2 molecules-29-00935-f002:**
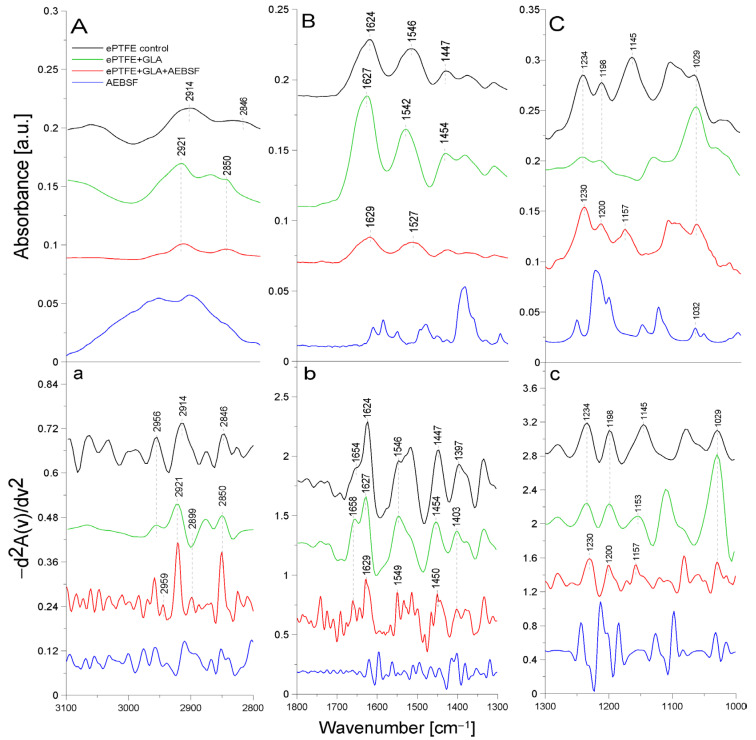
ATR-FTIR spectra of AEBSF: control ePTFE prosthesis (ePTFE control), ePTFE prosthesis functionalized with GLA (ePTFE + GLA), ePTFE prosthesis functionalized with GLA and AEBSF (ePTFE + GLA + AEBSF), AEBSF molecule alone (AEBSF); (**A**)—3100–2800 cm^−1^ region of ATR-FTIR spectra, (**B**)—1800–1300 cm^−1^ region of ATR-FTIR spectra, (**C**)—1300–1000 cm^−1^ region of ATR-FTIR spectra; (**a**–**c**)—inverted second derivatives of spectra presented in (**A**–**C**).

**Figure 3 molecules-29-00935-f003:**
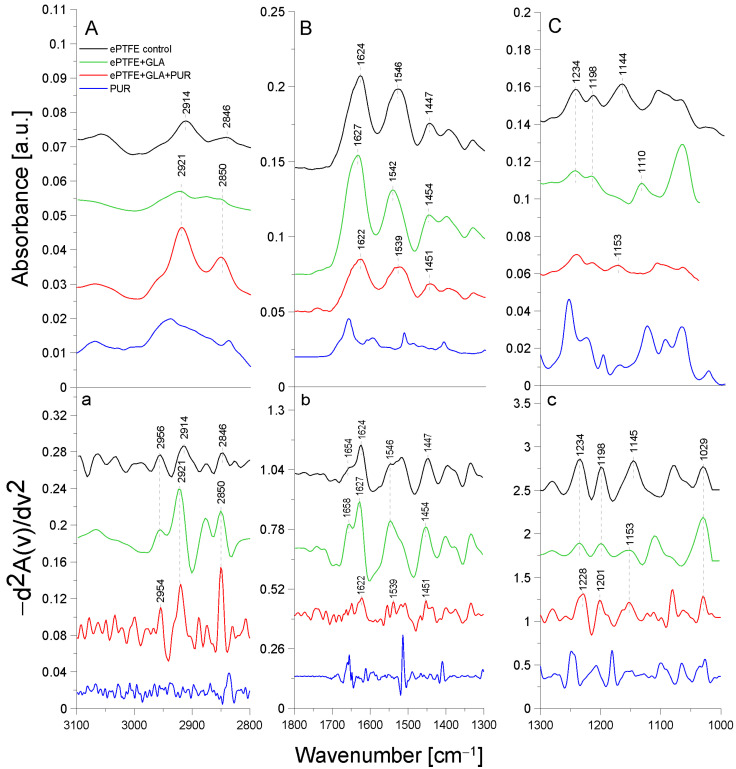
ATR-FTIR spectra of PUR: control ePTFE prosthesis (ePTFE control), ePTFE functionalized with GLA (ePTFE + GLA), ePTFE functionalized with GLA and PUR (ePTFE + GLA + PUR), PUR molecule alone (PUR); (**A**)—3100–2800 cm^−1^ region of ATR-FTIR spectra, (**B**)—1800–1300 cm^−1^ region of ATR-FTIR spectra, (**C**)—1300–1000 cm^−1^ region of ATR-FTIR spectra; (**a**–**c**)—inverted second derivatives of spectra presented in (**A**–**C**).

**Figure 4 molecules-29-00935-f004:**
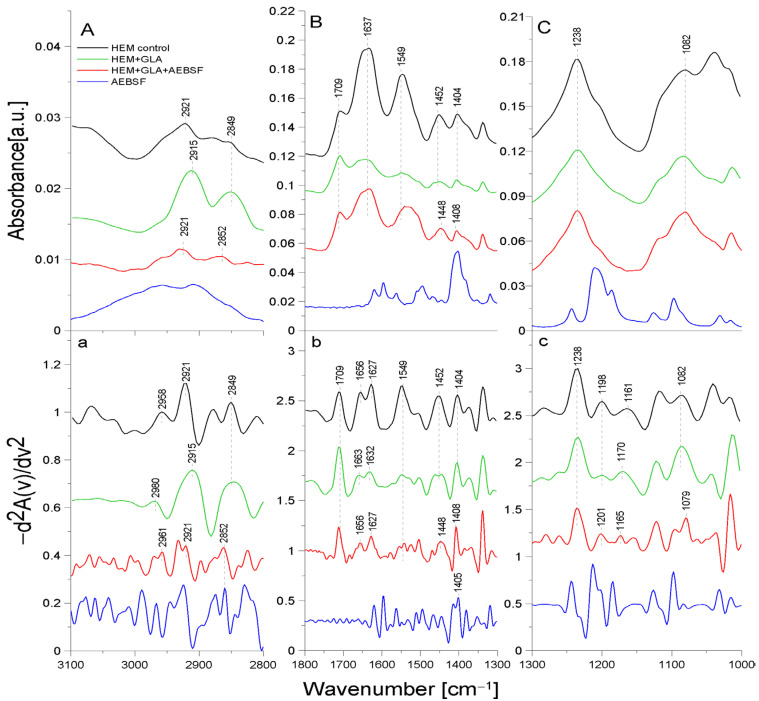
ATR-FTIR spectra of AEBSF: control HEM prosthesis (HEM control), HEM prosthesis functionalized with GLA (HEM + GLA), HEM prosthesis functionalized with GLA and AEBSF (HEM + GLA + AEBSF), AEBSF molecule alone (AEBSF); (**A**)—3100–2800 cm^−1^ region of ATR-FTIR spectra, (**B**)—1800–1300 cm^−1^ region of ATR-FTIR spectra, (**C**)—1300–1000 cm^−1^ region of ATR-FTIR spectra; (**a**–**c**)—inverted second derivatives of spectra presented in (**A**–**C**).

**Figure 5 molecules-29-00935-f005:**
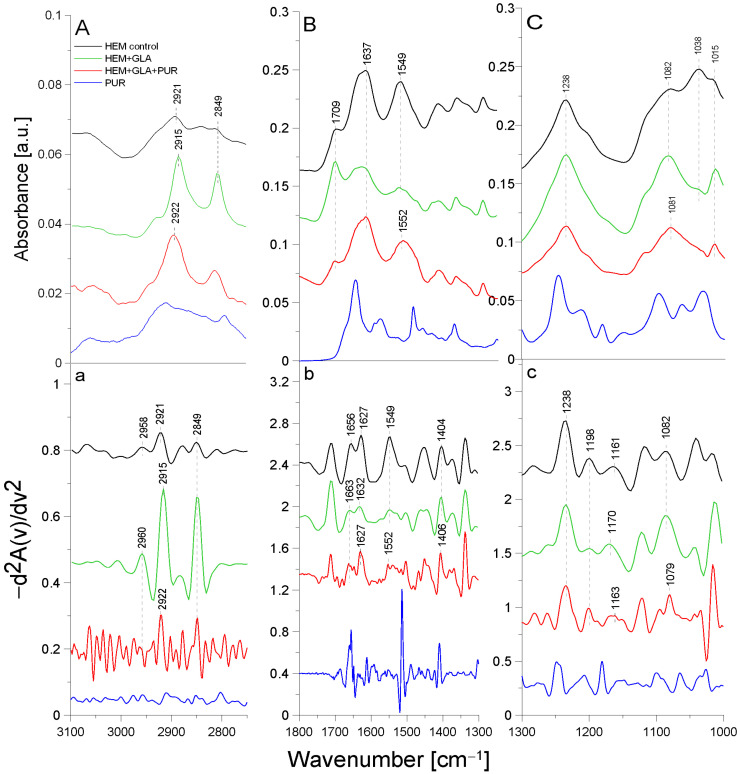
ATR-FTIR spectra of PUR: control HEM prosthesis (HEM control), HEM functionalized with GLA (HEM + GLA), HEM functionalized with GLA and PUR (HEM + GLA + PUR), PUR molecule alone (PUR); (**A**)—3100–2800 cm^−1^ region of ATR-FTIR spectra, (**B**)—1800–1300 cm^−1^ region of ATR-FTIR spectra, (**C**)—1300–1000 cm^−1^ region of ATR-FTIR spectra; (**a**–**c**)—inverted second derivatives of spectra presented in (**A**–**C**).

**Figure 6 molecules-29-00935-f006:**
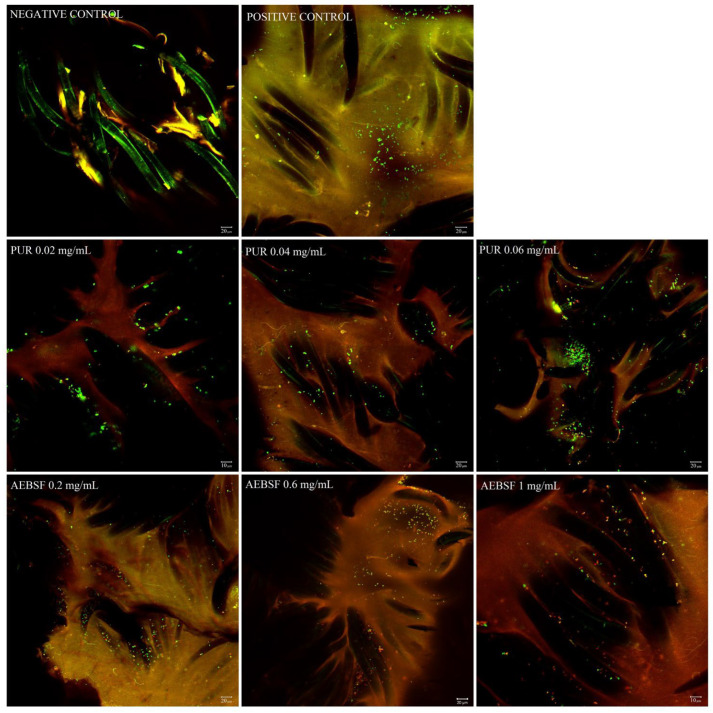
CLSM images of *Staphylococcus aureus* biofilm formed over 48 h on the HEM prosthesis treated with different concentrations of PUR (0.02 mg/mL, 0.04 mg/mL, 0.06 mg/mL) and AEBSF (0.2 mg/mL, 0.6 mg/mL, 1 mg/mL) for 3 h. Green fluorescence represents live bacteria stained with SYTO 9, whereas red fluorescence represents dead bacteria stained with propidium iodide. Negative control—fibers of the HEM prosthesis; positive control—*S. aureus* biofilm on the HEM prosthesis. Images were taken at 400× zoom for AEBSF 1 mg/mL and for PUR 0.02 mg/mL; other images were taken at 200× zoom.

**Figure 7 molecules-29-00935-f007:**
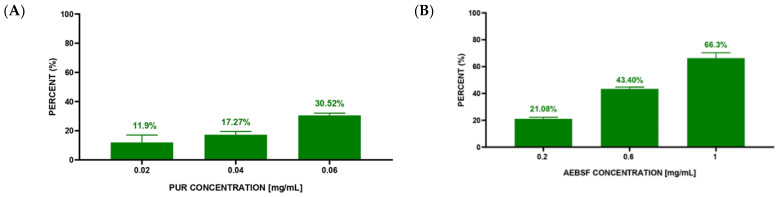
Mortality of *S. aureus* cells after 3 h of incubation with different concentrations of inhibitors—PUR (**A**) and AEBSF (**B**).

**Figure 8 molecules-29-00935-f008:**
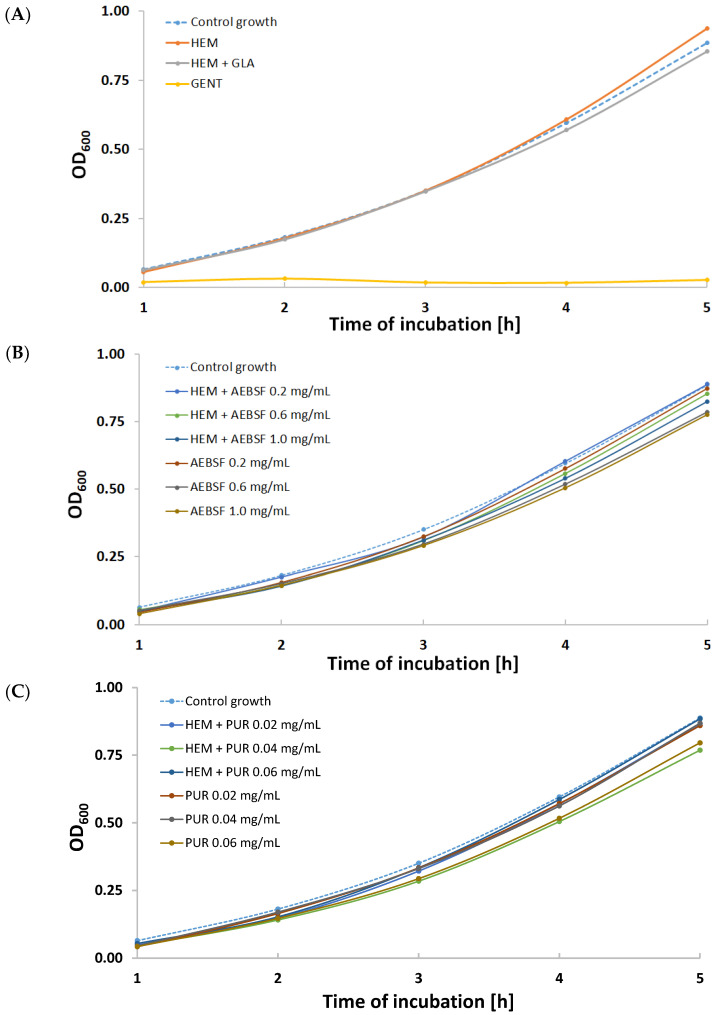
Growth curves for various variants of *S. aureus* suspension cultures incubated with the addition of (**A**) gentamicin (GENT), unmodified Hemagard prosthesis (HEM), and GLA-activated Hemagard prosthesis (HEM + GLA); (**B**) native (AEBSF) and immobilized AEBSF (HEM + AEBSF); (**C**) native (PUR) and immobilized puromycin (HEM + PUR).

**Figure 9 molecules-29-00935-f009:**
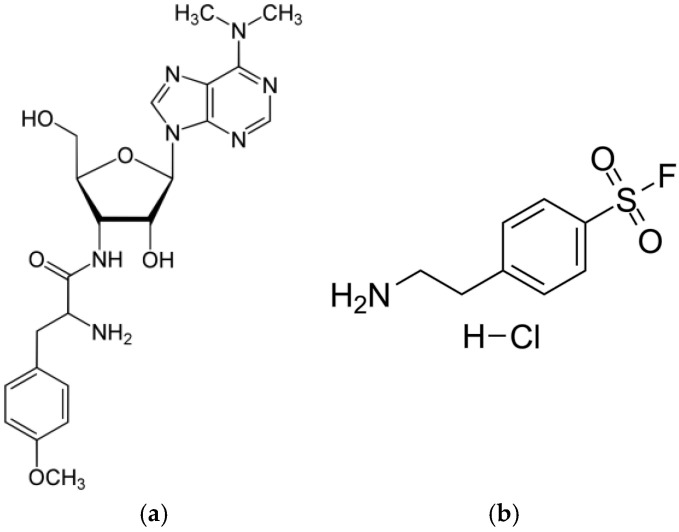
Structures of PUR (**a**) and AEBSF (**b**).

**Table 1 molecules-29-00935-t001:** Identification of ATR-FTIR bands in the spectra of TRYP, AEBSF, and PUR; str.—stretching; bend.—bending vibrations.

Band [cm^−1^]	Origin	Characteristic
1637; 1636	amide I group C=O (70–85%) C-N (10–20%) (β-helix)	protein region
1543; 1538; 1539	amide II group N-H bond bend. C-N bond str.	protein region
1656	aromatic ring vibrations	from PUR
1599	-C=C group str.	from AEBSF

**Table 2 molecules-29-00935-t002:** Identification of ATR-FTIR bands in ePTFE prosthesis spectra (control ePTFE and functionalized with GLA, AEBSF, and PUR); str.—stretching; asym.—asymmetric; sym.—symmetric; bend.—bending vibrations.

Band [cm^−1^]	Origin	Characteristic
2959; 2956; 2954	-CH group str.	polyethylene level
2921; 2914	-CH_2_ group sym.	polytetrafluoroethylene region
2850; 2846	-CH_2_CH_3_ group str.	polytetrafluoroethylene region
1654; 1658 1627; 1624; 1622; 1629	amide I group C=O bond str. (70–85%) C-N bond bend. (10–20%) N-H bond bend. (α-helix and β-helix)	protein (gelatin) region
1546; 1549; 1539	amide II group N-H bond bend. C-N bond str.	protein (gelatin) region
1447; 1454; 1450; 1451	-CH_2_	polytetrafluoroethylene region
1403; 1397	-CH_3_	carboxylate portion of polyethylene
1234; 1230; 1228	-CF_2_ group str.	polytetrafluoroethylene region
1200; 1201; 1198	-CF_2_ group asym. str.	polytetrafluoroethylene region
1145; 1153; 1157	-CF_2_ group sym. str.	polytetrafluoroethylene region
1029	-C-O and C-C str.	carbohydrate structure of gelatin

**Table 3 molecules-29-00935-t003:** Identification of ATR-FTIR bands in HEM prosthesis spectra (control HEM and functionalized with GLA, AEBSF, and PUR); str.—stretching; sym.—symmetric; def.—deformation; bend.—bending vibrations.

Band [cm^−1^]	Origin	Characteristic
2960; 2961; 2958	-CH group str.	from methylene group
2921; 2915; 2922	-CH_2_ group sym.	from polyester
2849; 2852	-CH_2_ group str.	from polyester
1709	-C=O group sym.	carbonyl group from polyester
1627; 1632; 1656; 1663	amide I group C=O bond str. (70–85%) N-H bond bend. (β-helix)	protein (gelatin) region
1549; 1552	amide II group N-H bond bend. C-N bond str.	protein (gelatin) region
1452; 1448	-CH group def.	from polyester
1404; 1405; 1406; 1408	-OH group	carboxylate portion of polyester fiber
1238; 1201; 1198; 1170; 1165; 1161; 1163; 1082; 1079	-CO group	carbonyl group in polyester chains

**Table 4 molecules-29-00935-t004:** The number of *S. aureus* cells after 5 h of incubation in various suspension culture variants. HEM—unmodified prosthesis; HEM + GLA—GLA-modified HEM prosthesis; HEM + ABSF—HEM prosthesis modified with AEBSF (0.2 mg/mL, 0.6 mg/mL, 1.0 mg/mL); AEBSF—solutions of native AEBSF inhibitor; HEM + PUR—HEM prosthesis modified with PUR (0.02 mg/mL, 0.04 mg/mL, 0.06 mg/mL); PUR—solutions of the native PUR inhibitor. Data are presented as average values obtained from three independent measurements (*n* = 3). The variants with the highest cell count were marked with bold.

Sample Type	Average Number of Bacteria [CFU/mL]
**Control growth**	**1.82 × 10^9^**
**HEM**	**1.93 × 10^9^**
HEM + GLA	1.77 × 10^9^
HEM + AEBSF 0.2 mg/mL	1.82 × 10^9^
HEM + AEBSF 0.6 mg/mL	1.76 × 10^9^
HEM + AEBSF 1.0 mg/mL	1.72 × 10^9^
AEBSF 0.2 mg/mL	1.77 × 10^9^
AEBSF 0.6 mg/mL	1.65 × 10^9^
AEBSF 1.0 mg/mL	1.60 × 10^9^
HEM + PUR 0.02 mg/mL	1.79 × 10^9^
HEM + PUR 0.04 mg/mL	1.60 × 10^9^
HEM + PUR 0.06 mg/mL	1.71 × 10^9^
PUR 0.02 mg/mL	1.74 × 10^9^
PUR 0.04 mg/mL	1.75 × 10^9^
PUR 0.06 mg/mL	1.63 × 10^9^

## Data Availability

The data presented in this study are available on request from the corresponding author.

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
