# Peer review of "Attachment of Proteolytic Enzyme Inhibitors to Vascular Prosthesis—An Analysis of Binding and Antimicrobial Properties"

_molecules, 2024, doi:10.3390/molecules29050935_

Round 1
Reviewer 1 Report
Comments and Suggestions for Authors
The research aims to prove interaction of trypsin and puromycin as well as AEBSF (synthetic serine protease inhibitor). Other question of the work is the inhibition of biofilm formation on vascular prostheses covered with protease inhibitor molecules.
The search of the interactions of trypsin is not new, but the main component of the trypsin involved in the binding is different in this work than in other literature. But trypsin is only a model for the enzyme inhibition activity measurement, and the result of Figure 1 is not a real proof.
Figure 2 represents the experiments to prove the binding of the different components to the prostheses. Detailed description of the spectra are given with conclusion of proper binding of GLA, AEBSF and PUR. Some of these reactions were shown in their previous works, this time different method is used with more details. Figure 4 shows the changes of the bound molecules after immobilization. I am not sure that these measurements gave any further knowledge to the molecules used for covering the vascular prostheses to protect against bacterial infections.
The authors carried out experiment using bacterial strains in their previous work with AEFBS serine protease inhibitor. This time they use only St. aureus, which gave the best result for them previously. Antibacterial effect was tested previously, this time they use another method to check the inhibition of biofilm formation.
In summary: the results gave no new information about the molecules used. The methods used in the work are new, the analyses are very detailed, and seems, that there are additional poofs for previous results. The conclusions of the results are correct, their usefulness are questionable. The quality of the figures are good, the literate cited is appropriate. Maybe the amount of information in this article is less than enough for publishing.
Author Response
Dear Reviewer,
on behalf of all manuscript authors, I thank you for the valuable comments and suggestions that will help us improve the quality of our manuscript. Below are our answers to your comments and suggestions:
The research aims to prove interaction of trypsin and puromycin as well as AEBSF (synthetic serine protease inhibitor). Other question of the work is the inhibition of biofilm formation on vascular prostheses covered with protease inhibitor molecules.
The search of the interactions of trypsin is not new, but the main component of the trypsin involved in the binding is different in this work than in other literature. But trypsin is only a model for the enzyme inhibition activity measurement, and the result of Figure 1 is not a real proof.
Figure 2 represents the experiments to prove the binding of the different components to the prostheses. Detailed description of the spectra are given with conclusion of proper binding of GLA, AEBSF and PUR. Some of these reactions were shown in their previous works, this time different method is used with more details. Figure 4 shows the changes of the bound molecules after immobilization. I am not sure that these measurements gave any further knowledge to the molecules used for covering the vascular prostheses to protect against bacterial infections.
The authors carried out experiment using bacterial strains in their previous work with AEFBS serine protease inhibitor. This time they use only St. aureus, which gave the best result for them previously. Antibacterial effect was tested previously, this time they use another method to check the inhibition of biofilm formation.
In summary: the results gave no new information about the molecules used. The methods used in the work are new, the analyses are very detailed, and seems, that there are additional poofs for previous results. The conclusions of the results are correct, their usefulness are questionable. The quality of the figures are good, the literate cited is appropriate. Maybe the amount of information in this article is less than enough for publishing.
Answer: Taking into account the suggestions of Reviewer 1 and also those included in Review 2, we decided to perform additional analyzes and experiments that will provide measurable evidence that the modified biomaterials have antimicrobial activities. A quantitative analysis of the mortality of S. aureus cells exposed to PUR and AEBSF molecules was made using CLSM analysis and obtained results were added to the new version of the manuscript (Figure 7). We also analyzed the inhibition of S. aureus growth in suspension cultures in contact with fragments of the modified Hemagard prosthesis and with native inhibitor molecules (a new subsection added to the work; lines 435-450 and 462-500).
We sincerely hope that adding the above-described data to the manuscript will bring new knowledge about the action of the inhibitors tested and will be hard evidence that PUR and AEBSF have the potential as antimicrobial agents.

Reviewer 2 Report
Comments and Suggestions for Authors
In this manuscript, the authors present a novel approach to combat prosthetic infections by covalently attaching antimicrobial molecules to different materials. After covalently linking puromycin and AEBSF to two prosthetic materials, the authors employed ATR-FTIR spectroscopy to confirm these covalent interactions by identifying changes in bonds.
Despite the thorough and clear report of the spectroscopy results, the antimicrobial activity section appears to be weak. Firstly, the methods indicate that biofilms were formed on the prosthesis before incubating the GLA-activated, biofilm-covered prosthesis with puromycin and AEBSF solution. This raises questions about differentiating the antimicrobial activity of free inhibitors versus covalently-linked ones on the prosthesis. The antimicrobial activity of puromycin and AEBSF in solution has been reported previously.
Secondly, the CLSM images don't seem to clearly demonstrate antimicrobial activity. The authors should quantify the images and perform statistical analyses. Given the significant background in the negative control and the potential of confocal microscopy for detecting single bacterium, the discrete round particles in the positive control images are more likely to be S. aureus, which remain mostly green except for the AEBSF 1mg/ml condition.
Moreover, considering the high background staining, the authors should use an additional method, such as CFU count before and after antimicrobial treatment, to validate the antimicrobial activity alongside CLSM.
Minor Comments:
1. Abbreviations should be defined at their first appearance in the manuscript, such as PUR, ATR-FTIR, TRYP, ePTFE, GLA, HEM.
2. It’s unclear to me why the interaction between trypsin and the other two inhibitors was examined. What’s the hypothesis here?
3. Please specify the magnification of the CLSM used by the authors in the methods.
Author Response
Dear Reviewer,
on behalf of all manuscript authors, I thank you for the valuable comments and suggestions that will help us improve the quality of our manuscript. Below are our answers to your comments and suggestions:
In this manuscript, the authors present a novel approach to combat prosthetic infections by covalently attaching antimicrobial molecules to different materials. After covalently linking puromycin and AEBSF to two prosthetic materials, the authors employed ATR-FTIR spectroscopy to confirm these covalent interactions by identifying changes in bonds.
Despite the thorough and clear report of the spectroscopy results, the antimicrobial activity section appears to be weak. Firstly, the methods indicate that biofilms were formed on the prosthesis before incubating the GLA-activated, biofilm-covered prosthesis with puromycin and AEBSF solution. This raises questions about differentiating the antimicrobial activity of free inhibitors versus covalently-linked ones on the prosthesis. The antimicrobial activity of puromycin and AEBSF in solution has been reported previously.
Answer: In response to doubts related to the effect of inhibitors immobilized on vascular prostheses compared to native inhibitors, we think that this issue needs to be addressed. Differences in action may occur because the inhibitor immobilized on the prosthesis does not have as much “freedom in action” compared to the native inhibitor, and access to the active site of the enzyme is more complicated. On the other hand, low-molecular inhibitors in their native form are not very stable, and the covalent immobilization on biomedical materials increases their stability. An additional experiment was done, and the prostheses with covalently immobilized AEBSF and PUR inhibitors were incubated with S. aureus suspension cultures (control samples were also prepared - prostheses without the immobilized inhibitor and GLA-activated prostheses). OD values of the bacterial suspension in contact with fragments of modified prostheses and native inhibitors were measured. In addition, the values of CFU per mL after five hours of incubation were calculated. These tests complemented the experiments carried out on a confocal microscope and confirmed the hypothesis that inhibitors of proteolytic enzymes may inhibit the multiplication and formation of S. aureus biofilm.
Secondly, the CLSM images don't seem to clearly demonstrate antimicrobial activity. The authors should quantify the images and perform statistical analyses. Given the significant background in the negative control and the potential of confocal microscopy for detecting single bacterium, the discrete round particles in the positive control images are more likely to be S. aureus, which remain mostly green except for the AEBSF 1mg/ml condition.
Answer: Unfortunately, the microscopic photos presented in this work are of poorer quality than the original full-resolution photos. Therefore, it is more difficult to notice differences in the antimicrobial activities of different variants of inhibitors used in the experiment. Thus, as suggested in the review, we performed a quantitative analysis using the ImageJ 1.53t program based on the obtained microscopic photos. The data were presented as a bar chart (made in GraphPad Prism 8) illustrating the percentage of S. aureus cell death (Figure 7). Analyzes were performed using three independent observations for each variant of inhibitor concentration (three repetitions), and the mean and standard deviation of the samples were calculated.
Moreover, considering the high background staining, the authors should use an additional method, such as CFU count before and after antimicrobial treatment, to validate the antimicrobial activity alongside CLSM.
Answer: Such quantitative analysis, along with determining the CFU/mL density, was performed following the reviewer's suggestion, and data were included in a new subsection of the manuscript (lines 435-450 and 462-500).
Minor Comments:
- Abbreviations should be defined at their first appearance in the manuscript, such as PUR, ATR-FTIR, TRYP, ePTFE, GLA, HEM.
Answer: The Abbreviations section has been added to the manuscript (after Keywords).
- It’s unclear to me why the interaction between trypsin and the other two inhibitors was examined. What’s the hypothesis here?
Answer: Two serine protease inhibitors were selected for analyzes - puromycin (PUR) and 4-(2-aminoethyl)benzenesulfonyl fluoride hydrochloride (AEBSF) to check their activities after binding to the enzyme active centre; in our work, it was trypsin (the model serine-type proteolytic enzyme). The research hypothesis assumes that both inhibitors can bind to the enzyme, effectively blocking its activity. Proteolytic enzymes play essential functions in all living organisms, including digestion, immune response, tissue regeneration, and blood coagulation; they are an ideal therapeutic target in combating pathogens.
Sharifloo and co-workers [2017] analyzed trypsin’s activity from the intestine of caterpillar larvae (Pieris brassicae L.). They observed a significant decrease in this enzyme's activity after using an AEBSF inhibitor in vitro. Kudryakova and co-workers [2019] checked the inhibition of serine protease-L1 activity after exposure to AEBSF inhibitor. They used trypsin as a positive control and observed the inhibition of its activity after 10 minutes of inhibitor addition.
In our research, we wanted to confirm the previously described antimicrobial properties of serine protease inhibitors tested and find a correlation between them and the biophysical tests performed (ATR-FTIR spectroscopy), assessing the structural changes occurring after interactions between both compounds (the enzyme and the inhibitor). The results obtained confirm our research hypothesis.
- Sharifloo A., Zibaee A., Sendi J.J., Jahroumi K.T, Biochemical characterization a digestive trypsin in the midgut of large cabbage white bufferly, Pieris brassicae L. Bulletin of Entomological Research 108(4) (2018), 501-509.
- Kudryakova I.V., Gabdulkhakov A.G., Tishchenko S.V., Afoshin A.S., Vasilyeva N.V., Serine bacteriolytic protease L1 of Lysobacter XL1 complexed with protease inhibitor AEBSF: features of interaction, Process Biochemistry 80 (2019), 89-94.
- Please specify the magnification of the CLSM used by the authors in the methods (lines 457-548).
Answer: The information about the magnification has been added to section 3.7. Confocal laser scanning microscopy analysis (CLSM).
Round 2
Reviewer 1 Report
Comments and Suggestions for Authors
The authors added new experiments which made their results more convincing.
I accept the paper in this form for publication.
Reviewer 2 Report
Comments and Suggestions for Authors
The authors have addresses all of my comments.